# Navigating in Virtual Environments: Does a Map or a Map-Based Description Presented Beforehand Help?

**DOI:** 10.3390/brainsci11060773

**Published:** 2021-06-10

**Authors:** Chiara Meneghetti, Francesca Pazzaglia

**Affiliations:** 1Department of General Psychology, University of Padova, 35131 Padova, Italy; francesca.pazzaglia@unipd.it; 2Interuniversity Research Center in Environmental Psychology (CIRPA), 00185 Rome, Italy

**Keywords:** navigation, virtual environment, map, map-based description, individual visuospatial differences

## Abstract

Background. One of the aims of research in spatial cognition is to examine the factors capable of optimizing environment learning from navigation, which can be examined using a virtual environment (VE). Different learning conditions can play an important part. Aim. This study examined the benefits of presenting configured information (layout with elements arranged in it) using a map or verbal description before a learner navigates in a new environment. Method. Ninety participants were assigned to three learning groups of 30 individuals (15 males and 15 females). Before participants navigated in a VE, one group was shown a map of the environment (“map before navigation”), a second group read a map-like description of the environment (“description before navigation”), and a third group started navigating without any prior input (“only navigation”). Participants then learned a path in a VE (presented as if they were driving a car). Their recall was subsequently tested using three types of task: (i) route retracing; (ii) pointing; (iii) path drawing. Several measures were administered to assess participants’ individual visuospatial and verbal factors. Results. There were no differences between the three groups in route retracing. The “map before navigation” group performed better than the “only navigation” group in both the pointing and the path drawing tasks, however, and also outperformed the “description before navigation” group in the path drawing task. Some relations emerged between participants’ individual difference factors and their recall performance. Conclusions. In learning from navigation, seeing a map beforehand benefits learning accuracy. Recall performance is also supported, at least in part, by individual visuospatial and verbal factors.

## 1. Introduction

### 1.1. Navigating in a Virtual Environment

Navigating in an environment is a crucial everyday activity. It is a complex process by means of which we experience a space from an egocentric point of view, based on sensorimotor information about our position in space, self-to-object distances, and self-motion. Navigating enables us to learn a series of landmarks, turns, and changes of direction, and to memorize a set of place–action associations [1]. Learning spatial information by navigating in an environment gives rise to a mental representation, or cognitive map [2], that can be externalized to accomplish various environmental demands, such as repeating a previously learned route, estimating distances and directions, or drawing a map [3].

Studies on navigation can benefit from using virtual environments (VE), which can serve as a good approximation of the real world [4] when exploring mechanisms and strategies underlying navigation, and the properties of the resulting mental representations. They also offer several advantages, such as an effective control over the visual content, and an easy and accurate collection of behavioral data [5,6]. The validity of a VE increases when it reproduces three-dimensional entities in fully immersive conditions that resemble real-life sensorimotor experiences, with real-time interaction [7], and the sense of “being there” [8]. The mental representations deriving from exploring VEs can have properties similar to those of real navigation [4], though they differ in the amount of cognitive resources required (which is greater when VE is used [9]), and in some underlying mechanisms (such as the brain network involved [10]).

Using VEs can help to shed light on which conditions optimize learning from navigation, such as whether the formation of mental representations is best served by spatial information presented before or during a navigation experience.

In considering how information is presented to people learning from navigation, there is also the matter of the frame of reference, i.e., the coordinate system relative to which spatial information is encoded and represented in memory [11]. Spatial information can be processed and memorized using the egocentric frame, in which we perceive elements (such as landmarks) in the environment in relation to our own body (self-to-object relations), or using the allocentric frame, in which such elements are perceived in relation to one another (object-to-object relations). It is well known that whether we acquire the same environmental information from looking at a map (i.e., stressing the allocentric frame) or from navigating (i.e., stressing the egocentric one) affects the features of the mental representations we form. Studies found that participants learning an environment from a map performed better in tasks assessing their configured knowledge (such as map drawing), whereas participants who had moved around in the same environment performed better in navigation tasks [12,13]. The pattern of results was similar when navigation was experienced in a VE [4,14,15].

When we are navigating, we learn a path sequentially, as seen from our own, egocentric point of view (or what we could also call navigation-based information). Our resulting mental representation may have egocentric properties that can be externalized, for instance, when we repeat the same path (a task that retains the same view as in the learning phase). Our mental representation may also incorporate allocentric (or what we could also call map-based) properties, however, that can be externalized when we consider landmarks in relation to one another, as to find a shortcut or draw a map of the environment, for instance. Combining navigation-based and map-based information can help us to obtain good-quality mental representations that incorporate multiple views [16]. Several variables influence whether or not our mental representations of environments learned from navigation will also incorporate map-based knowledge [17]. One way to help people form mental representations from navigating experiences, that can incorporate not only navigation-based as well as and map-based knowledge, is to provide configured information about the environment in question (a general layout with elements arranged in it) to facilitate the formation object-to-object knowledge after navigating in it.

The main aim of the present study was to see whether presenting an environment as a whole (using visual or verbal media) before people navigate in a VE can help improve the mental representation they form of it, enabling it to incorporate navigation-based as well as map-based knowledge. Most studies in this line of research focused on presenting information during a navigation exercise, while few did so beforehand. A review of the former nonetheless offers insight on the latter condition, which is considered in the present study.

### 1.2. Spatial Information Provided during Navigation

While navigating in an environment, we often use navigation aids, such as a map or a GPS system, to find our way. Studies examining the mental representations we form of environments when what we learn from navigating is matched with other types of environmental information have shown that maps have a positive impact on our ability to learn from navigation. Ishikawa et al. [18] compared three conditions for people exploring a new environment: two with some form of visual navigation aid (a map showing the whole layout and all the elements it contained or a GPS showing a partial view with the elements appearing in relation to the observer’s position); the third with none. They found that participants who used the GPS travelled longer distances and stopped more often during their navigation experience than those who used the map or those who relied on direct experience alone. In subsequent wayfinding and map drawing tasks testing participants’ recall of the environment learned, GPS users performed less well than participants using no navigation aids. This inspiring study led to numerous others examining which type of navigation aid works best during navigation [19,20]. The latest study by Münzer et al. [21] showed that seeing an environment (as on a map) while navigating benefits self-to-object knowledge (as needed to judge relative directions in pointing tasks) and object-to-object knowledge (as measured with configurational tasks); this was influenced by the alignment of the map vis-à-vis the heading during navigation. This finding supports the idea that providing map-based information during an egocentrically based experience like navigation favors an interplay between the frames of reference (egocentric and allocentric), resulting in a mental representation that incorporates multiple types of spatial knowledge. There are also reports of how a map’s features affect people’s mental representations. Seeing maps that emphasize configured features (showing a whole layout with elements such as landmarks and streets) while navigating makes us produce mental representations with better map-like features (as measured with map drawing tasks) than seeing a map that accentuates local features along the way and decision points, which are more associated with a route-like knowledge [22,23,24].

Another way to provide spatial information while someone is navigating in an environment is to use language. In everyday navigation we commonly rely on verbal instructions while moving towards our destination. This involves using a multimedia learning approach [25,26] and, according to the dual coding theory [27], presenting the same information using visual and verbal formats helps us to learn better than from a single format. Presenting verbal and visual media limits the cognitive load [28] because it enables us to distribute our resources better than when we receive a large amount of the same type of information (visuospatial, for instance); this is because we have processing systems that enable us to handle multiple types of information at the same time [29].

It is well known that encoding environmental information from verbal descriptions is associated with the formation of mental representations with spatial properties [30] that, to some extent, resemble the representations formed using maps [31] or navigating [32]. Descriptions can also convey configured information about an environment (i.e., how landmarks are located in relation to one another in a layout) using cardinal points (e.g., “to the north”, or “on the southern side”). This type of information is called a survey description (what we have called map-based), and it differs from a route description, which presents a path step by step, using egocentric words (“go straight on”) [33]. The use of map-based (or survey) descriptions is associated with a better recall of information with map-based features (as tested using map drawing tasks, for instance) than when route descriptions are used [12,34,35]. Some studies examined the role of different characteristics of verbal input provided during navigation. Saucer et al. [36] and Hund and Minarik [37] compared the role of spatial instructions given during navigation using landmarks as reference points (landmark descriptors) or cardinal points (cardinal descriptors). Overall, their results showed that when participants (all participants in Hund and Minarik [37], or only males in Saucer et al. [36]) used cardinal descriptors they navigated in the environment faster and more accurately than when they used landmark descriptors. This finding supports the conviction that conveying information verbally based on cardinal points, which provides information on the global layout of an environment, facilitates learning from navigation and improves the resulting mental representation.

Other studies combined different types of navigation aid (verbal vs. visual) used during navigation to examine their impact on navigation efficacy and spatial knowledge acquisition [38,39,40,41]. The results are mixed and need to be contextualized in relation to how the visual and verbal media were used and the recall tasks administered. In fact, verbal instructions seemed to have a more beneficial effect than map presentations while people were driving in an unfamiliar environment [42] or forming environmental knowledge [38]. Parush and Berman [40] found an initial advantage of using verbal instructions (lists of steps to reach a target from a learner’s location), measured in terms of the time taken and the number of steps. Later, however, the advantage of using a map while learning from navigation emerged when their participants were asked to navigate without any aids, and when they performed orientation (pointing) tasks. That said, there seems to be a positive impact on environment knowledge, however, when both verbal and visual media are used to provide configured information (layouts and elements inside them). Krukar et al. [39] asked participants to learn a path in a city using different types of instructions: (i) focused on local elements (encountered one after the other); or (ii) on “chunks” of route information (spatial chunking); or (iii) presenting a route in relation to a global context (orientation). The instructions given could be only verbal or matched with visual input (with features similar to the verbal instructions). The results showed that providing the third type of instructions accompanied by a map was associated with a better landmark recall and sketch map performance, compared with the other types of instructions. The second and third types of verbal instructions, which both present the whole route to be learned, were associated with a better performance than the first.

Overall the results of studies associating visual or/and verbal media with navigation showed that (i) using a map (showing the layout and the elements in it) during navigation improves navigation efficiency and environment knowledge acquisition [18,21,23]; (ii) verbal instructions based on cardinal points improve performance during navigation [36,37] and environment knowledge [39]; (iii) when maps and verbal media are compared, verbal media seem to better orient navigation [40,42], but when both convey configured information, they both have a positive impact on environment knowledge [39]. In discussing the results, some of these studies attributed the positive effect of verbal instructions given during navigation to the lesser workload involved in processing both verbal and visuospatial information at the same time [38,39]. This is based on the assumption that both verbal and visuospatial working memory are involved when verbal and visuospatial information is processed, distributing the burden by comparison with having to process a greater amount of information of the same nature [29]. One way to reduce the cognitive load experienced during navigation could be to see configured global information about an environment (the layout and the elements in it) before navigating in it. Presenting map-based information before a navigation experience could provide a mental outline in which to place environmental information subsequently gleaned step by step as the navigation experience proceeds.

### 1.3. Spatial Information Provided before Navigation

There is evidence in the literature of the benefit of presenting spatial information before navigation using both visuospatial and verbal media.

Seeing a map before navigating in a new environment has a positive impact on recall task performance [43,44]. Meilinger et al. [45] tried to see how allocentric and egocentric frames work together in learning from navigation. They gave participants a chance to learn about an environment in one of four ways: from navigating in a VE; from looking at a map; from looking at the map and then navigating in the VE; or from navigating in the VE and then looking at the map. The results showed that learning about the environment from either a map or navigation led to a better pointing performance (fewer errors) when the imaginary view in the pointing task was aligned with the orientation of the map or the path navigated, respectively. When both modalities (map and navigation) were used, their influence depended on the order in which they were used. The first modality used influenced the features of the mental representation: pointing performance was related more to its map-based features when the map was shown first, and more to its navigation-based features when the navigation experience came first. This indicated that the (first or only) modality used to convey environmental information has an impact on its mental representation (in line with [13]). Stites et al. [46] recently showed that the effect of presenting a map (with various features, simple 2D and 3D, complex 2D) before or during navigation is modulated by individuals’ sense of direction. In particular, when individuals with a good sense of direction studied a simple map before navigating, or used it while navigating, their spatial performance improved (they made fewer pointing and distance estimating errors). On the other hand, individuals with a poor sense of direction performed better without a map than with a map. This would mean that individual differences can have a role in modulating any effects of manipulating the availability of additional external input (before or during navigation) (see also [37]).

Other studies found that the features of people’s final mental representations of an environment change when their navigation experience is preceded by verbal instructions such as the assignment of a spatial goal that focuses their attention on the global layout (spatial relations between landmarks) or on moves (self-to-object relations) [12,47,48]. Taylor et al. [12] gave participants an egocentrically oriented goal (i.e., they were asked to focus on the route) or an allocentric-based goal (i.e., they were asked to focus on the layout of a building) before they explored an environment (the floor of a building) by actually navigating in it or looking at a map. The results showed that the type of information provided beforehand influenced their environment learning: an egocentrically oriented goal enhanced their performance in egocentric-based tasks, while an allocentric-oriented goal enhanced performance in map-based tasks. Whatever the goal, however, participants performed better in allocentric-based tasks if they had studied a map, or in egocentric-based tasks if they had navigated in the building (in line with [3]). More recently, Meneghetti et al. [48] asked participants to learn a preset path in a VE on a desktop PC, after instructing them to focus either on the path (an egocentrically oriented goal) or on finding a shortcut (an allocentric-based goal that makes it necessary to consider multiple relations between landmarks). The results showed that participants given egocentric-based instructions performed better in a route repeating task than in a shortcut finding task; those given map-based instructions were better at finding shortcuts than in repeating a route. In addition, individual working memory capacity supported performance in the navigation task (route repeating), indicating that individual differences contribute to modulating navigation performance when different goals are set (see also [14]). This line of research is certainly interesting. It suggests that providing map-based verbal information before a navigation experience gives rise to an environment representation with map-like properties. The verbal instructions used in these studies suggested a goal, however; they did not provide a description of the overall environment (its layout and the arrangement of the elements it contained), as done in the studies providing descriptive verbal instructions while participants were navigating [36,37,39]. In short, the role of map-based descriptions presented before a navigation experience has yet to be established.

Reports on the use of different types of input (maps or verbal instructions) before navigation are limited. Morett et al. [49] presented participants with a map or descriptive (egocentrically based) verbal instructions, or both simultaneously, or first the map and then the verbal description, or vice versa, before a navigation experience that simulated driving. The results showed that being presented with both types of input at the same time was associated with a greater accuracy in a navigation task, supporting the multimedia theory that having access to multiple (visual and verbal) media is beneficial [25,26]. The study also showed that participants performed better in terms of navigation skills (identifying novel routes) when they were shown a map first. This would suggest that a map presented earlier enabled them to better contextualize the information they acquired from navigation. It should be noted that providing a map or verbal description before a navigation task did not necessarily improve performance. Schlender et al. [50], for instance, found that navigation performance benefited when information was provided during navigation but not if it was provided beforehand. Path-finding performance was better with any kind of navigational cues (used before or during navigation) than with none, however.

Overall, these studies indicate that presenting a map [45] or a verbally conveyed map-based goal [12,48] before a navigation experience prompts environment representations with map-like properties, and there is some evidence to indicate that seeing a map before navigating enlarges our spatial knowledge [49]. Our ability to manage different types of navigation-related input seems to be modulated by individual differences [14,46,48].

Seeing configured information about an environment (its layout and how elements are arranged in it) before navigating in it therefore gives us a chance to better contextualize the information we subsequently acquire while navigating, enabling an interplay between both (egocentric and allocentric) spatial frames of reference. To our knowledge, however, the utility of seeing a configured layout of an environment before navigating in it, and the role of the format used—visual (a map) or verbal (a description)—has not been systematically examined.

### 1.4. Rationale and Aim of the Study

The main aim of the present study was to examine whether presenting configured information (i.e., a layout and a whole path within it) of an environment before navigation—using different media, i.e., a map or a map-based verbal description—influences spatial learning efficacy and the features of the resulting mental representation.

Most of the literature has focused on spatial information offered during navigation (navigation aids), and suggests that presenting map-based information, using either a map [18,21] or verbal instructions based on cardinal directions—that, to a certain extent, provide information about a configured environment [36,37]—influences people’s mental representations, facilitating their navigation and promoting spatial (map-based) knowledge [23]. To a lesser extent, other studies examined the role of presenting configured information before a navigation experience, showing an impact on spatial learning, with evidence that maps [45] and verbal instructions—allocentric-based goal [12]—both promote map-based knowledge. The effect of presenting map-based information (conveyed by maps or verbal instructions) before a navigation experience has yet to be analyzed, however. Examining the impact of configured information (conveyed by maps or verbal instructions) on learning from navigation enables us to investigate the interplay between allocentric and egocentric frames of reference [11], and between the multiple presentation formats (verbal and visual) typical of the multimedia approach [25,26].

Therefore, it will be examined whether presenting configured information (i.e., a layout and a whole path within it) before navigation (step by step along a path, from an egocentric point of view) facilitated the construction of a mental representation able to include navigation-based as well as map-based knowledge. We examined whether the media used to present this information beforehand (a map or a verbal map-based description) made a difference to the final recall performance.

As a complementary aim, given that learning modalities can be related to individual differences [46,48], and the role of the latter in learning from navigation is well known [3,9,51], we examined the relation between measures of individual differences (in terms of abilities and self-reports) and environment learning accuracy as a function of learning condition and type of recall task.

To accomplish these aims we planned a study that involved learning a path in a fully immersive VE resembling real-life navigation [6]. The path chosen included only a street with turns, without any landmarks or other environmental features, given the influence of navigation performance (e.g., [52,53]) and the brain network involved in generating cognitive maps [54] (see also [55,56]).

The learning from navigation involved three conditions: before navigating in the VE, one group of participants, called the “map before navigation” group, looked at a map (showing a layout with paths, and highlighting the path to be navigated); another, called the “description before navigation” group, read a map-based (survey) description of the environment (its layout and the main direction of the path to be navigated); a third, called the “only navigation” group, started navigating without any information. After a guided navigation phase, participants performed a series of recall tasks. In a first, navigation-based task, they were asked to retrace the path they had learned previously. Then there were two tasks—pointing and path drawing—that involved the ability to manage information in their mental representations; the latter task is especially reliant on the representation having map-like features. Participants completed several visuospatial measures assessing a subset of abilities, including the Mental Rotations Test [57], and working memory tasks with visuospatial and verbal spans. They were also administered self-report measures: a Sense of Direction and Spatial Representation scale [58]; a Spatial Anxiety scale [59].

We expected the following results:(i)having access to configured (map-based) information before navigating (an egocentrically oriented activity) can prompt a better mental representation of an environment compared with navigating alone because this would mean having both map-based and navigation-based frames of reference available during the learning phase. This benefit of having map-based information in advance is supported by: studies on the use of aids during navigation, such as a map [18] or directions based on cardinal points [37]; studies in which map-based information was received before navigating (be it a map [45] or a map-based goal [12]). We examined whether using different media (visual vs. verbal) affected learning from navigation differently. Here different results are possible. If the influence of the prior configured (map-based) information prevails over that of the navigation experience (whatever the format used), we might expect a benefit of both visual (map) and verbal (map-based description) media. On the other hand, if the format used is important, then visual or verbal media could have a different impact on navigation learning. We might reasonably expect a map to work better than verbal presentation because it enables the context of the navigation experience to be visualized (as previously suggested [45,49]), using different complementary frames of reference [11]. Alternatively, we might expect a verbal description to work better than a map as this would be a complementary way to convey the same information (in line with the multimedia approach [25,26]), giving rise to a lesser cognitive load as information of a different nature would need to be processed. This has been suggested in studies examining visual versus verbal information used during navigation [38,40,42]. In line with these expectations, we will examine whether presenting configured information (a map or map-based description) beforehand prompts a better environment learning performance compared with navigation alone. Differences in recall task accuracy between our three groups of participants will be examined, also as a function of the type of recall task. Knowing that presenting map-based information seems to prompt mental representations with configuration-like features (as suggested by studies examining the benefits of such information during navigation [39,40] or beforehand [45,48]), it was worth examining whether this benefit emerges particularly in a path drawing task, or in a pointing task that demands a change of view;(ii)accuracy in learning from navigation should relate to measures of individual differences [3,51]. We examined whether the role of individual differences changes in relation to learning condition, with visuospatial abilities [48] and preferences [46] possibly helping people to manage information (both when presented earlier and/or in multiple formats). Any differences as a function of recall tasks were also investigated.

## 2. Materials and Methods

### 2.1. Participants

The present study was approved by the Ethics Committee for Research in Psychology at the University of Padova. It involved a total of 90 undergraduates (45 females and 45 males) from the University of Padova, with a mean age of 23.18 (SD = 1.85). Participants were randomly assigned to one of three learning groups: “map before navigation”, “description before navigation”, “only navigation”, with 30 in each group, 15 males and 15 females. Considering the sample as a whole, 74.4% (*n* = 67) played video games for at least 1 h a week, and 88.9% (*n* = 80) were regular drivers. A power analysis (using G Power) hypothesizing a medium effect size (η^2^ = 0.30) on 3 (learning condition) × 3 (recall tasks) MONAVA indicated that 81 participants sufficed to find main effects of learning conditions on the three tasks (power = 0.80).

### 2.2. Materials

#### 2.2.1. Learning from Navigation

Learning phase. A virtual path of an open-air environment was prepared in route view using the TileRacer software, a freeware 3D racing game (http://tileracer.model-view.com, accessed on 25 May 2009). The path was set in a hilly environment with distinctive paved roads and greenery on boards and all around. No landmarks were included. The path consisted of 9 turns at a 90° angle (5 to the right and 4 to the left). It was projected on a 17″ computer screen placed 50 cm away from the participant. The path was covered from a starting point to a destination using a car virtually driven by the participant and moving at a speed of 45 km/h. Figure 1 shows an example of how the path appeared to the driver. To move within the environment, participants used a steering wheel and pedal system (Momo Racing Force Feedback). The steering wheel was on the table in front of the screen and was used to turn right and left. The pedals (one to brake and one to move forward) were on the floor under the table. A helmet (V8 Head Mount Display) was used for full immersion in the VE. A different path with similar environmental features was used for practice.

Information provided before navigation. The map shows the layout of the environment with a compass, and the intersecting roads. It also shows the path to cover, highlighted in red (Figure 2a). The description (75 words long) presents the layout and the main directions taken by the path based on cardinal points (Figure 2b).

Anxiety question. Participants were asked: “How much anxiety did you experience while navigating?” and their answer was given using a 5-point Likert scale (from 1 = “not at all” to 5 = “very much”).

#### 2.2.2. Recall Tasks

Route retracing task. This involves covering the same path, using the steering wheel, pedals and helmet. When participants took a wrong turn or missed a turn, the experimenter stopped them, and took them back to their position before they made a mistake. Then they drove on. The sum of their errors (for each wrong direction) was recorded (max 9 errors).

Pointing task. This consists in imagining being at the arrival point and indicating the direction of the starting point. A circle with an arrow pointing straight up from the center to the circumference was used to represent the direction in which the participant was facing. They indicated the direction of the starting point by tracing another line from the center to the boundary of the circle. Their score was given by the degrees of discrepancy between their answer and the correct direction (max 180° degrees of error).

Path drawing task. This consists in drawing the path covered on a blank sheet of paper. Participants earned one point for each turn positioned in the right order and the right direction; they lost one point for each turn positioned in the wrong direction or for each unnecessary turn (max 9 correct turns). The sheet of paper used to draw the path was also divided into four quadrants and participants scored one point if their path arrived in the top right-hand quadrant (i.e., to the north-east).

#### 2.2.3. Individual Difference Measures

Corsi Blocks task—backward version (adapted; [60]). This consists of a wooden board with 9 cube-shaped blocks on it, and the experimenter taps increasingly long sequences of blocks (from 2 to 8 blocks, with two sequences for each length). Participants are asked to reproduce the sequences in reverse order. The task is terminated when participants fail to repeat both sequences of the same length. The score was the maximum number of blocks correctly recalled (max 8).

Digit span task—backward version [61]. This consists in repeating sequences of digits (previously presented verbally by the experimenter) in reverse order. The length of the sequences ranged from 2 to 8 (with two sequences for each length). The final score was the longest sequence successfully recalled (max 8).

Mental Rotations Test (MRT, [57]). This measures the ability to rotate three-dimensional objects in space. There are 20 items, each comprising one 3D target figure (assembled cubes) and four options that differ in degrees of rotation or are mirror images. The task consists in identifying the two figures that match the target figure but are in a rotated position. The maximum time allowed to complete the task is 8 min. The score is the sum of correct answers (1 point for correctly identifying both figures in each item, max 20). Cronbach’s α for the present sample was 0.81.

Sense of Direction and Spatial Representation Scale (SDSR, [58]). This comprises 11 items measuring three factors: general sense of direction and preference for survey mode; knowledge and use of cardinal points; preference for landmark-centered route mode. Participants’ degree of agreement with the statements was expressed on a Likert scale from 1 (not at all) to 5 (very much). The score is the sum of the items relating to each factor. Cronbach’s α for the present sample was 0.77, 0.79, 0.61.

Spatial Anxiety Scale (SAS, [59]). This comprises 8 items measuring anxiety experienced in wayfinding situations. The degree of anxiety was expressed on a Likert scale from 1 (not at all) to 6 (very much), and the sum of the ratings was calculated. Cronbach’s α for the present sample was 0.77.

### 2.3. Procedure

Participants attended an individual session at the VR lab at Padova University’s General Psychology Department. They completed the individual difference measures first (Corsi Blocks and Digit Span tasks, MRT, SDSR and SAS, in counterbalanced order across participants). Other measures not included in the present study were also administered. Then participants were randomly assigned to one of three groups. All participants started with a practice session, wearing the helmet for a fully immersive navigation exercise in a VE. They sat in front of the computer screen and were instructed to use the steering wheel to turn right or left and the two pedals to brake or move forward. The experimenter used another computer screen placed alongside the one used by the participants to monitor their performance during the navigation. For practice purposes, participants could freely navigate a path without being given any directions or time limit. The practice session was stopped when a participant was able to drive the car forward and turn to left and right. Then participants in the “map before navigation” group were shown the map and those in the “description before navigation” group were given the description to read (for around one minute), while those in the “only navigation” group were given no additional information. The navigation session started, and participants moved along the path following the experimenter’s verbal instructions (turn right, turn left) when they came to a crossroads until they reached the destination. Immediately afterwards, participants completed the route retracing, pointing, and path drawing tasks, in that order. Finally, participants answered the question about their level of anxiety experienced during the navigation exercise. A summary of the materials and procedure is given in Figure 3.

## 3. Results

### 3.1. Preliminary Analyses

The level of anxiety experienced while navigating was quite low, M = 2.31 (SD = 0.93), and no significant differences were found by group or gender (all F values < 1). A multivariate ANOVA was carried out on the scores and ratings obtained in the working memory tasks and MRT, SDSR and SAS, with group and gender (males vs. females) as between-participant variables. Gender was included because of its impact on visuospatial individual difference measures [51]. The results showed no significant effect of group, F (14,158) = 1.40, *p* = 0.15), but there was a significant effect of gender, F (7, 78) = 3.11, η^2^ = 0.22, *p* = 0.006. Univariate ANOVAs showed significant effects in the MRT, F (1, 88) = 13.01, η^2^ = 0.13, *p* = 0.001, where males (M = 11.91 SD = 4.76) scored higher than females (M = 8.24 SD = 4.89); in the use of cardinal points F (1, 88) = 6.60, η^2^ = 0.07, *p* = 0.02, where males (M = 8.39 SD = 3.56) gave higher ratings than females (M = 6.54 SD = 3.24).

### 3.2. Effect of Group on Task Performance

The scores were normally distributed in all tasks in the three groups given the asymmetry and kurtosis were mainly in the range of −1/+1. The range of values were as follows: asymmetry −1.14/0.85 and kurtosis −0.76/0.54 in the “map before navigation” group; asymmetry −0.22/0.28 and kurtosis −1.41/−0.89 in the “description before navigation” group and asymmetry −0.02/0.25 and kurtosis −0.80/−0.71 in the “only navigation” group.

A multivariate ANOVA was carried out, considering route retracing (errors), pointing task (degrees of error) and path drawing task (accuracy) as dependent variables, and gender [51] and group as independent variables. The results showed a significant effect of group, F (6, 166) = 9.04, η^2^ = 0.25, *p* ≤ 0.001, but not of gender (F < 1). The impact of group on the three recall tasks was examined separately with three univariate ANOVAs. A main effect of group was found for the pointing task, F (2, 87) = 5.23, η^2^ = 0.11, *p* = 0.007, and the path drawing task, F (2, 87) = 24.97, η^2^ = 0.37, *p* ≤ 0.001, but not for the route retracing task (F (2, 87) = 1.81 *p* = 0.17; mean errors M = 4.28 SD = 2.08 for the three groups). The effects of group remained when, as a control, the amount of time spent on video games and driving experience were input as covariates in the ANOVAs. Table 1 shows the means and standard deviations for the three tasks in the three groups. Post-hoc comparisons—considering those with *p* ≤ 0.01 as significant—showed that the “map before navigation” group (M = 32.40 SD = 33.46) had fewer degrees of error in the pointing task than the “only navigation” group (M = 61.97 SD = 35.84; *p* = 0.007), while the “description before navigation” group’s performance (M = 44.43 SD = 37.40) did not differ significantly from that of the other two groups. In the path drawing task, the “map before navigation” group (M = 6.70 SD = 2.83) performed better than the “only navigation” group (M = 0.30 SD = 4.64) or the “description before navigation” group (M = 1.06 SD = 3.79; *p*_s_ ≤ 0.001), while the latter two did not differ significantly. Given that the asymmetry and kurtosis values were slightly below −1 in two cases, we transformed the scores on a logarithmic scale (the log-transformed scores are reported in Appendix A) and ran the same analyses again to confirm the results obtained. The results completely overlapped those already reported.

When we looked at how many participants completed their path drawing in the right (top right-hand) quadrant, there was a significant difference between the groups, χ^2^ (2) = 19.43, *p* ≤ 0.001. In the “map before navigation” group, 29 participants drew a path that arrived in the right quadrant (only one failed to do so), while this was true of 23 participants in the “description before navigation” group (seven arrived elsewhere), and 14 in the “only navigation” group (16 reached another quadrant).

### 3.3. Relations between Individual Visuospatial Difference Measures and Recall Task Performance in the Three Groups

Pearson’s correlations between the individual visuospatial difference measures and the three groups’ scores in the recall tasks were calculated. No significant correlations emerged (considering r values associated with *p* ≤ 0.001 at least). That said, the highest correlation values (corresponding to a medium to large effect size) were as follows: the preference for a landmark-route mode was moderately high for path drawing accuracy, r = −0.449 (*p* = 0.013), in the “description before navigation” group; the digit span score was moderately high for route retracing errors, r = −0.464 (*p* = 0.01), in the “only navigation” group. The correlations between all individual visuospatial difference measures and recall task performance in the three groups are reported in the Appendix A.

Stepwise regression models were run to clarify the role of individual differences as a function of group and recall task. In the “description before navigation” group, path drawing (F (1, 28) = 7.03 *p* = 0.013) performance was predicted by a preference for a landmark-route mode (β = −0.449; *p* = 0.013), and the model explained 17% of the variance (R^2^ adjusted = 0.17). In the “only navigation group”, route retracing errors (F (2, 27) = 7.019 *p* = 0.004) were predicted by digit span scores (β = −0.505; t = −3.214, *p* = 0.03), and by SAS ratings (β = 0.358; t = −2.77, *p* = 0.031), and the model explained 29% of the variance (R^2^ adjusted = 0.29).

## 4. Discussion

The main aim of the present study was to examine whether configured information presented visually (map) or verbally (a map-based written description) before navigation in a VE benefited the spatial knowledge acquired. Three groups of participants navigated in a VE in fully immersive conditions as if they were driving a car. One group (“only navigation”) did so without any prior knowledge of the environment, one (“description before navigation”) read a description of the path to be taken, and one (“map before navigation”) saw a map before starting the navigation exercise. Route retracing, pointing and path drawing tasks were administered after participants had completed their virtual journey. Several measures of individual differences were also completed and the performance of participants in the three groups was examined in relation to these measures.

The results showed an effect of group in relation to recall task performance. The “map before navigation” group performed best. Having seen a map before navigating, produced a more successful performance than in the “only navigation” group in both pointing (fewer degrees of error) and path drawing (accuracy). The “map before navigation” group also performed better in the path drawing task than the “description navigation” group. The three groups did not differ in route retracing performance, however, as they all made a moderate number of mistakes (around 4.28; max 9). These results indicate that, while advance knowledge did not benefit performance in a task similar to the one experienced in the learning phase (route retracing after learning from navigation), it did help in tasks that involved managing the newly acquired environmental knowledge to adopt a different imaginary view (pointing task) or draw the path travelled in a map-like view (path drawing task). A specific benefit was apparent when the environment had been presented earlier on a map. In other words, seeing the whole layout and path to cover enabled participants to construct a mental representation of the environment that gave them access to information not strictly related to the egocentric view they experienced during navigation, i.e., they acquired spatial knowledge that enabled them to adopt a different imaginary view (as measured with the pointing task) or a bird’s-eye view (measured with the path drawing task) of the environment they had navigated in. This finding supports the idea that creating the conditions for people to access both egocentric and allocentric frames of reference [11] by giving them a map (to acquire map-based information) and an experience of navigation (navigation-based) enables them to form a mental representation that incorporates multiple types of spatial knowledge. This is consistent with studies showing the benefit of using a map both during navigation [18,21] and beforehand [45]. They also support reports that maps better prompt the acquisition of map-based knowledge, confirming that maps enable the acquisition of configured environmental knowledge [13], even when associated with learning from navigation [23,45,49].

The beneficial effect of reading descriptive information before navigating is limited. The performance of the “description before navigation” group was halfway between that of the other two groups in the pointing task, and in the path drawing task participants in the “description before navigation” group reached the correct quadrant of the final destination more often than those in the “only navigation” group. In other words, verbally conveyed map-based information did little to improve recall task performance in our experimental conditions. This contrasts with findings of a beneficial effect of verbal information before [12,48] and during navigation [36,37], and even—to a certain extent—of verbal being superior to visual material (when both media were used; [38,39]), in line with multimedia theory [25,26]. Our results indicate that, although language conveying configured information can be effective during [36,37] and before a navigation experience [12,48], a map is more powerful in promoting spatial knowledge acquisition [49].

Our analysis on the role of individual difference factors showed that they had a limited role in relation to group and recall task. Here the comment of the relationships between variables associated at a moderate level. In our “only navigation” group, a better verbal working memory (digit span) was associated with fewer route retracing errors. This supports the suggestion that, when environmental knowledge is gained visually—probably stressing the visuospatial working memory system [62]—then verbal working memory capacity supports accuracy in learning from navigation [48]. Further, in the “only navigation” group a greater spatial anxiety also seemed to have a role in increasing the number of route retracing errors (with a moderate degree of correlation). This finding of a role for spatial anxiety in limiting learning from navigation is also consistent with the literature [3,51], and deserves to be better examined. In our “description before navigation” group, a stronger preference for a landmark-route mode (or egocentric view) was associated with a less accurate path drawing; performance in this task seemed to be facilitated by an individual preference for a survey (map-based) mode [35].

Although we found gender-related differences (in favor of males) in mental rotation and in the knowledge and use of cardinal points (in line with previous studies, [51]), there was no clear evidence of individual differences in working memory [48] or sense of direction [46], for instance, having a role as a function of group. More evidence is needed on this issue, also because of the limited number of relations found here, unlike studies examining individual differences in the abilities involved in learning from navigation [9,51].

To sum up, the results of the present study showed that seeing a map before navigation led to the formation of a better environment representation than navigating alone (as tested with pointing and path drawing tasks), while reading a description of the environment was of more limited benefit. Individual difference factors had little influence on learning from navigation. This superior value of a map seen before navigating is certainly interesting, but more research needs to be done, exploiting the potential offered by VE. The VE technique achieves a more well-developed 3D visualization not only in spatial cognition but also in other related disciplines, such as multimedia cartography [63,64], which uses VE to reproduce maps and environments that can be experienced from an egocentric perspective in real time to obtain additional information not experienced with real physical exploration. Advances in VE technologies in multiple domains facilitate the development of maps and environments in fully immersive virtual settings to corroborate the positive effect of maps, for instance, as this is not always confirmed [50], and to take their features into account [21,39]. Their comparison with verbal media should also be expanded, working on the type of content—from goal-setting instructions [12] to detailed map-based information [39,49]. To clarify the optimal timing of the relation between map-based and navigation-based information, it would also be worth comparing the effects of using the same type of media (maps or descriptions) before as well as during navigation [49]. It should be noted that earlier research in which information was given before and/or during navigation included landmarks in several cases [40,41,42,43,44,45], and their effects in better orienting navigation and favoring the representation were largely confirmed [52,53], with specific brain networks anchoring the landmarks to the cognitive map [54,55,56]. In further studies it would therefore be worth considering the effects of including landmarks along paths and in environments navigated—in relation to learning condition (maps or descriptions presented before or during navigation). Another indicator of navigation ability to consider concerns the moves made during a navigation experience (even the initial one), as done in studies on navigation aids [20,21]. Considering this measure as well (not only tasks after navigation) could generate additional information on navigation ability.

Extending this line of research will allow us to expand on theoretical knowledge of how allocentric and egocentric frames of reference interact [11], and how different media (visual vs. verbal) can be integrated [25,26]. Having identified the ideal learning conditions, the potential benefit could be applied in various ways. One such application could be as follows: starting from the promising evidence that VE locomotion and navigation training produces benefits on environment representation (in estimating distance [65], for instance, or a distance covered [66]; see also [67,68] for earlier works), it would be interesting to deliver navigation-based training that systematically offers a preview of map-based information (using visual and/or verbal input), then assess the effects of repeated experience on representation accuracy. Such a learning condition might benefit a participant’s ability to manage information from different views. This could be of interest to the whole population, but especially to specific populations that have difficulty managing and integrating navigation and map-based information of the same environment, such as older adults [69], and individuals with atypical development [70]. Another application could involve designing and interfacing systems for orienting ourselves in the environment. This would enable new interfacing systems to be used when navigating, integrating information provided not only during the navigation (as in GPS use; [20,21]) but also beforehand. The implementation of such devices capable of presenting multiple types of information and views can be tested and developed exploiting the potential of VE technologies. This points to new challenges in our approach to devices and interface systems [71,72] with a view to optimizing navigation in different conditions.

## 5. Conclusions

To conclude, this study offers fresh evidence of the beneficial effect of seeing configured information (a layout with elements arranged in it) on a map before learning from navigation, as it facilitates the construction of spatial knowledge. It goes to show how VE can be used to simulate the best conditions in which to learn from navigation, and the resulting mental representation of the environment encountered.

## Figures and Tables

**Figure 1 brainsci-11-00773-f001:**
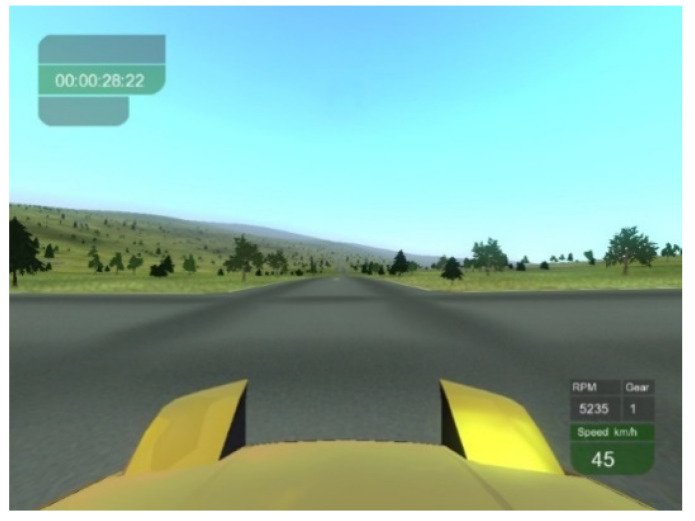
A typical scene as a driver neared a crossroads.

**Figure 2 brainsci-11-00773-f002:**
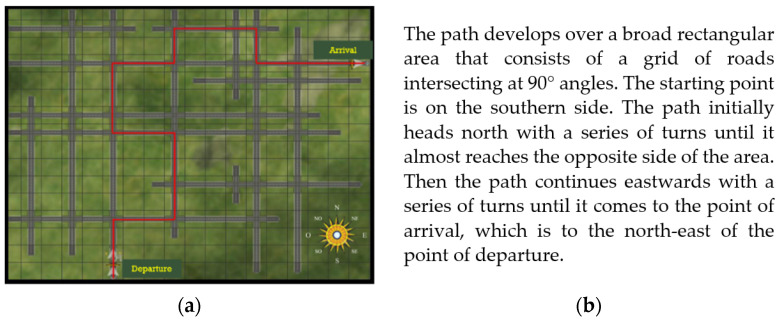
(**a**) The map studied by the “map before navigation” group; (**b**) the description of the environment read by the “description before navigation” group.

**Figure 3 brainsci-11-00773-f003:**
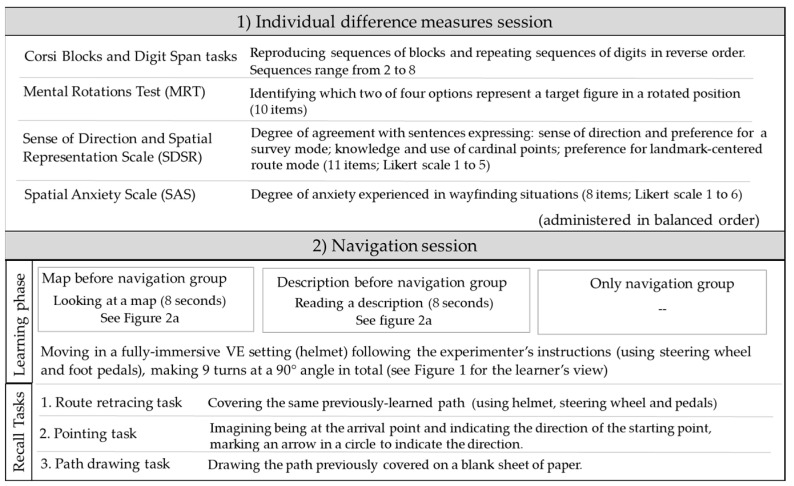
Summary of materials and procedure.

**Table 1 brainsci-11-00773-t001:** Means and standard deviations (in brackets) of performance in route retracing (number of errors), pointing (degrees of error) and path drawing (number of correct turns) by group.

	Map before Navigation Group	Description before Navigation Group	Only Navigation Group
Route retracing task	4.36 (1.79)	3.73 (1.91)	4.73 (2.42)
Pointing task	32.40 (33.46)	44.43 (37.40)	61.97 (35.83)
Path drawing task	6.70 (2.83)	0.30 (4.64)	1.07 (3.79)

## Data Availability

The data presented in this study are available on request from the corresponding author.

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
