# Peer review of "Navigating in Virtual Environments: Does a Map or a Map-Based Description Presented Beforehand Help?"

_brainsci, 2021, doi:10.3390/brainsci11060773_

Round 1
Reviewer 1 Report
Meneghetti & Pazzaglia examine the benefits presenting information visually (in the form of a map) or verbally (in the form of a map-based verbal description) before a person navigates a new environment. The conclude that they can improve the accuracy of route retracing, pointing to the direction of the starting point, and drawing the path that was covered during the navigation.
While some of the results are interesting, ai have a number of concerns about this manuscript.
- In the introduction, the experiment, and the discussion, the authors largely ignore the use of landmarks, even though they play a crucial role in navigation (see e.g., Epstein, Patal, Julian, & Spiers, Nature Neuroscience 2017, 20(11), 1504-1513 and many of the papers cited there). While at this point, it is not possible to include landmark-based navigation in the experiment, a thorough discussion is required in the paper.
- While the authors discuss the effect of group (map before navigation, description before navigation, no information before navigation) on a large number of variables, the effect on the navigation session, i.e. the number of errors during navigation, is not presented. This needs to be added.
- I am very concerned that highly skewed error data (judged from the means and standard deviations of errors in Table 1) may bias the results of the Anovas and paired comparisons. I would want to see, possibly in the Supplementary Material, how a normalizing transform affects the statistical analysis and results, and in case of large discrepancies, a thorough discussion is required.
- In the supplementary material, 63 correlations are presented with 2 being significant. An adjustment of the significance level using Bonferroni’s correction is required to keep type I errors under control.
Minor errors
- Line 468: What is Fs?
- Line 472: df’s of the F-test are missing
- Line 472 and later: Use 2 as an exponent in η2
- Line 486: Make clear that M=4.28 is the overall mean of all groups.
- Supplementary material table column labels: retracing nor reteacing
Author Response
Dear Reviewer,
Concerning our manuscript entitled “Navigating in virtual environments: does a map or a map-based description presented beforehand help?”, we have revised the paper following your suggestions. All changes are highlighted in red. An English mother-tongue professional translator had already checked the first version of the manuscript, and the parts changed or added in the second version.
- In the introduction, the experiment, and the discussion, the authors largely ignore the use of landmarks, even though they play a crucial role in navigation (see e.g., Epstein, Patal, Julian, & Spiers, Nature Neuroscience 2017, 20(11), 1504-1513 and many of the papers cited there). While at this point, it is not possible to include landmark-based navigation in the experiment, a thorough discussion is required in the paper.
Answer: We agree that landmarks play a relevant role in navigation and in its final accuracy. In introducing the experiment, we have specified that we focused only on navigation (or orientation) in an environment that did not include any landmarks, given their effects on the accuracy of final representations (we have added specific references) and on the brain network (as you underscored with the reference you mentioned) (see lines 307-310). In the Discussion we have added the consideration that further studies should better examine the effect of learning condition in a VE that includes landmarks, given their effects on mental representations (see lines 630-636).
- While the authors discuss the effect of group (map before navigation, description before navigation, no information before navigation) on a large number of variables, the effect on the navigation session, i.e. the number of errors during navigation, is not presented. This needs to be added.
Answer: The number of navigation errors was recorded only after participants followed the experimenter’s instructions, i.e. after the guided navigation. We did not record the errors made while following the instructions during the first navigation experience. Although our main interest was in the effect of learning condition on recall task performance after learning from navigation, which included navigation errors (with route retracing), it would also have been interesting to consider the number of errors made while learning from navigation (including the initial navigation experience). In the Discussion we have now mentioned that further studies should also consider performance during learning from navigation, as this was not done in our study (see lines 636-640).
- I am very concerned that highly skewed error data (judged from the means and standard deviations of errors in Table 1) may bias the results of the Anovas and paired comparisons. I would want to see, possibly in the Supplementary Material, how a normalizing transform affects the statistical analysis and results, and in case of large discrepancies, a thorough discussion is required.
Answer: Thank you for this suggestion. The asymmetry and kurtosis of each dependent variable (retracing, pointing, path drawing) in each group were mainly within the range of -1/+1, indicating that the scores were normally distributed (see lines 486-490). It is important to bear in mind that negative values were possible in the path drawing score (because one point was subtracted for each turn in the wrong direction or unnecessary turn), so the values near 0 in Table 1 do not indicate a floor effect.
However, since the scores were slightly below -1 (-1.14, -1.41) in two cases, we followed your suggestion and transformed the scores on a logarithmic scale. We ran the same analyses (MANOVAs and univariate ANOVAs) and the results completely overlapped with those already reported. We have added this information in the Results (see lines 509-512), and reported the descriptive statistics in the supplementary material (Table S1), but prefer not to describe them in detail in the Results (given the complete overlap with those already reported).
- In the supplementary material, 63 correlations are presented with 2 being significant. An adjustment of the significance level using Bonferroni’s correction is required to keep type I errors under control.
Answer: Thank you for this comment. Using Bonferroni's correction we should have considered p ≤.001, which corresponds to values of r >.55. The most correlated measures are r=-.449 and r=-.464, which are moderate to large in terms of effect size. We think these correlations are worth commenting on. We have now stated in the text that, while they are not significant measures, they do express a moderate-to-large strength of the relations (see lines 524-529). This better reported in in the Supplementary material. This is better specified in the discussion too (see lines 592-593).
Minor errors
- Line 468: What is Fs?
Answer: It means all F values considered. To avoid confusion, we have changed this to “all F values < 1”
- Line 472: df’s of the F-test are missing
Answer: The df has now been added (see line 480), also on line 498.
- Line 472 and later: Use 2 as an exponent in η2
Answer: We have corrected this and all other occurrences of η2..
- Line 486: Make clear that M=4.28 is the overall mean of all groups.
Answer: We have corrected this by writing: “mean errors M = 4.28, SD = 2.08 for the three groups”
- Supplementary material table column labels: retracing nor reteacing
Answer: Thank you. We have corrected this typo.
Reviewer 2 Report
The manuscript is concentrated on an experimental investigation in a virtual environment addressing a navigation tasks with three different conditions: map before navigation, description before navigation, only navigation. The main result reported is that the learning performance is increased if participants see a map before navigating through a virtual environment.
The manuscript appears in a mature state (icluding a clear discussion section which is hardly the case in review round 1), and the topic itself is timely. Before recommending a publication of this manuscript, I would like to mention the following points which the authors should reflect for a revised manuscript version:
- The topic of map-based navigation in IVEs is a topic which is not only considered in research of cognitive psychology and spatial cognition, but also in related disciplines of 3D visualization. Tu underline the importance of the interplay between maps and navigation in a 3D scenario (including a change in perspective to a first-person-view), the authors could also add some references where techniques and developments are included and discussed in the background section, see for e.g.: https://doi.org/10.3390/ijgi10020096 and https://doi.org/10.1007/s42489-019-00030-2
- Your three study conditions are related to training which is a topic which has regularly been discussed in research including spatial cognition tasks. The training with a map and a map-like description might likely have an impact on the navigation performance. For instance, it was recently shown that training regarding locomtion tools in immersive virtual environments (IVEs) increases the accuracy in distance estimations, a measure which is also related to navigation performance (see https://doi.org/10.3390/ijgi10030150 ). Earlier studies also provided evidence that training has an impact on spatial performance measures in IVEs, see for e.g. https://doi.org/10.1002/acp.1140 and https://doi.org/10.1002/acp.1174
Your approach of using (at least in two condition) a spatial training device to improve the navigation performance is interesting. Could you discuss the (potential) impact of training, even if you connot measure and quantify the impact afterwards?
- Could you please include a figure containing your study materials (including conditions), and a figure that visually sums up your procedure?
Author Response
Dear Reviewer,
Concerning our manuscript entitled “Navigating in virtual environments: does a map or a map-based description presented beforehand help?”, we have revised the paper following your suggestions. All changes are highlighted in red. An English mother-tongue professional translator had already checked the first version of the manuscript, and the parts changed or added in the second version.
The manuscript appears in a mature state (icluding a clear discussion section which is hardly the case in review round 1), and the topic itself is timely. Before recommending a publication of this manuscript, I would like to mention the following points which the authors should reflect for a revised manuscript version:
- The topic of map-based navigation in IVEs is a topic which is not only considered in research of cognitive psychology and spatial cognition, but also in related disciplines of 3D visualization. Tu underline the importance of the interplay between maps and navigation in a 3D scenario (including a change in perspective to a first-person-view), the authors could also add some references where techniques and developments are included and discussed in the background section, see for e.g.: https://doi.org/10.3390/ijgi10020096 and https://doi.org/10.1007/s42489-019-00030-2
Answer: Thank you for the suggestion. We have included these references offered by the multimedia cartography approach to support the potential of using VE to reproduce 3D scenarios in map and navigation views. This technological advance enables the development of maps and environments in a fully-immersive virtual setting for further research on the contribution of maps before and/or during navigation. See lines 618-623.
- Your three study conditions are related to training which is a topic which has regularly been discussed in research including spatial cognition tasks. The training with a map and a map-like description might likely have an impact on the navigation performance. For instance, it was recently shown that training regarding locomtion tools in immersive virtual environments (IVEs) increases the accuracy in distance estimations, a measure which is also related to navigation performance (see https://doi.org/10.3390/ijgi10030150). Earlier studies also provided evidence that training has an impact on spatial performance measures in IVEs, see for e.g. https://doi.org/10.1002/acp.1140 and https://doi.org/10.1002/acp.1174
Answer: Thank you for this suggestion. In the Discussion, we have now explained that, starting from the evidence of benefits of VE locomotion and navigation training on environment representation (and we have cited the works that you suggest and an additional one*), we have mentioned the possibility of considering navigation-based training that incorporates the presentation of map-based information to assess its benefit on environment representation. See lines 644-651.
* McLaren-Gradinaru, M., Burles, F.,Dhillon, I., Retsinas, A., Umiltà, A., Hannah J, Dolhan K & Iaria G. (2020) A novel training program to improve human spatial orientation: preliminary findings. Frontiers in Human Neuroscience, 14: 5. http://doi: 10.3389/fnhum.2020.00005
- Your approach of using (at least in two condition) a spatial training device to improve the navigation performance is interesting. Could you discuss the (potential) impact of training, even if you connot measure and quantify the impact afterwards?
Answer: We have added some expected results of navigation training in improving representation accuracy (see lines 650-651), and better explained its potential application when developing devices, such as interface systems, to approach multiple types of information and views (see lines 655-660).
- Could you please include a figure containing your study materials (including conditions), and a figure that visually sums up your procedure?
Answer: A Figure 3 has now been added to illustrate the materials and procedure.